# Coresets for Classification – Simplified and Strengthened

**Tung Mai**
Adobe Research
tumai@adobe.com

**Cameron Musco**
University of Massachusetts Amherst
cmusco@cs.umass.edu

**Anup Rao**
Adobe Research
anuprao@adobe.com

## Abstract

We give relative error coresets for training linear classifiers with a broad class of loss functions, including the logistic loss and hinge loss. Our construction achieves $(1 \pm \epsilon)$ relative error with $\tilde{O}(d \cdot \mu_y(X)^2/\epsilon^2)$ points, where $\mu_y(X)$ is a natural complexity measure of the data matrix $X \in \mathbb{R}^{n \times d}$ and label vector $y \in \{-1, 1\}^n$, introduced in [MSSW18]. Our result is based on subsampling data points with probabilities proportional to their $\ell_1$ *Lewis weights*. It significantly improves on existing theoretical bounds and performs well in practice, outperforming uniform subsampling along with other importance sampling methods. Our sampling distribution does not depend on the labels, so can be used for active learning. It also does not depend on the specific loss function, so a single coreset can be used in multiple training scenarios.

## 1 Introduction

Coresets are an important tool in scalable machine learning. Given $n$ data points and some objective function, we seek to select a subset of $m \ll n$ data points such that minimizing the objective function on those points (possibly where selected points are weighted non-uniformly) will yield a near minimizer over the full dataset. Coresets have been applied to problems ranging from clustering [HPM04, FL11], to principal component analysis [CEM+15, FSS20], to linear regression [DMM06, DDH+09, CWW19], to kernel density estimation [PT20], and beyond [AHPV05, BLK17, SS18].

We study coresets for *linear classification*. Given a data matrix $X \in \mathbb{R}^{n \times d}$, with $i^{th}$ row $x_i$ and a label vector $y \in \{-1, 1\}^n$, the goal is to compute $\beta^* = \arg\min_{\beta \in \mathbb{R}^d} L(\beta)$, where $L(\beta) = \sum_{i=1}^n f(\langle x_i, \beta \rangle \cdot y_i)$ for a classification loss function $f$, such as the logistic loss $f(z) = \ln(1 + e^{-z})$ used in logistic regression or hinge loss $f(z) = \max(0, 1 - z)$ used in soft-margin SVMs.

We seek to select a subset of $m \ll n$ points $x_{i_1}, \ldots, x_{i_m}$ along with a corresponding set of weights $w_1, \ldots, w_m$ such that, for some small $\epsilon > 0$ and all $\beta \in \mathbb{R}^d$,

$$\left| \sum_{j=1}^m w_j \cdot f(\langle x_{i_j}, \beta \rangle \cdot y_{i_j}) - L(\beta) \right| \le \epsilon \cdot L(\beta). \tag{1}$$

This *relative error coreset guarantee* ensures that if $\tilde{\beta} \in \mathbb{R}^d$ is computed to be the minimizer of the weighted loss over our $m$ selected points, then $L(\tilde{\beta}) \le \frac{1+\epsilon}{1-\epsilon} \cdot L(\beta^*)$.

It is well known that common classification loss functions such as the log and hinge losses *do not admit relative error coresets with $o(n)$ points* [MSSW18]. To address this issue, Munteanu et al. introduce a natural notion of the complexity of the matrix $X$ and label vector $y$, which we will also use to parameterize our results:

**Definition 1** (Classification Complexity Measure [MSSW18]). *For any $X \in \mathbb{R}^{n \times d}$, $y \in \{-1, 1\}^n$, let $\mu_y(X) = \sup_{\beta \neq 0} \frac{\|(D_y X \beta)^+\|_1}{\|(D_y X \beta)^-\|_1}$, where $D_y \in \mathbb{R}^{n \times n}$ is a diagonal matrix with $y$ as its diagonal, and $(D_y X \beta)^+$ and $(D_y X \beta)^-$ denote the set of positive and negative entries in $D_y X \beta$.*

Roughly, $\mu_y(X)$ is large when there is some parameter vector $\beta \in \mathbb{R}^d$ that produces significant imbalance between correctly classified and misclassified points. This can occur e.g., when the data is exactly separable. However, as argued in [MSSW18], we typically expect $\mu_y(X)$ to be small.

## 1.1 Our Results

Our main result, formally stated in Corollary 9, is that sampling $\tilde{O}\left(\frac{d \cdot \mu_y(X)^2}{\epsilon^2}\right)$ points according to the $\ell_1$ *Lewis weights* of $X$ and reweighting appropriately, yields a relative error coreset satisfying (1) for the logistic loss, the hinge loss, and generally a broad class of 'hinge-like' losses. This significantly improves the previous state-of-the-art using the same $\mu_y(X)$ parameterization, which was $\tilde{O}\left(\frac{d^3 \cdot \mu_y(X)^3}{\epsilon^4}\right)$ [MSSW18]. See Table 1 for a detailed comparison with prior work.

**Theoretical Approach.** The Lewis weights are a measure of the importance of rows in $X$, originally designed to sample rows in order to preserve $\|X\beta\|_1$ for any $\beta \in \mathbb{R}^d$ [CP15]. They can be viewed as an $\ell_1$ generalization of the *leverage scores* which are used in applications where one seeks to preserve $\|X\beta\|_2$ [CLM+15]. Like the leverage scores, the Lewis weights can be approximated very efficiently, in $\tilde{O}(\text{nnz}(X) + d^\omega)$ time where $\omega \approx 2.37$ is the constant of fast matrix multiplication. They can also be approximated in streaming and online settings [BDM+20]. Our coreset constructions directly inherit these computational properties.

The $\ell_1$ Lewis weights are a natural sampling distribution for hinge-like loss functions, including the logistic loss, hinge loss, and the ReLU. These functions grow approximately linearly for positive $z$, but asymptote at 0 for negative $z$. Thus, ignoring some technical details, it can be shown that $\sum_{i=1}^n f(\langle x_i, \beta \rangle \cdot y_i)$ concentrates only better under sampling than $\sum_{i=1}^n |\langle x_i, \beta \rangle \cdot y_i| = \|D_y X \beta\|_1$.

As shown by Cohen and Peng [CP15], taking $\tilde{O}(d/\epsilon^2)$ samples according to the Lewis weights of $X$ (which are the same as those of $D_y X$) suffices to approximate $\|D_y X \beta\|_1$ for all $\beta \in \mathbb{R}^d$ up to $(1 \pm \epsilon)$ relative error. We show in Thm. 8 using contraction bounds for Rademacher averages that it in turn suffices to approximate $\sum_{i=1}^n f(\langle x_i, \beta \rangle \cdot y_i)$ up to additive error roughly $\epsilon(\|X\|_1 + n)$. We then simply show in Corollaries 6 and 9 that by setting $\epsilon' = \Theta(\epsilon/\mu_y(X))$ and applying Def. 1, this result yields a relative error coreset for a broad class of hinge-like loss functions including the ReLU, the log loss, and the hinge loss.

| Samples | Error | Loss | Assumptions | Distribution | Ref. |
|---|---|---|---|---|---|
| $\tilde{O}\left(\frac{d \cdot \mu(X)^2}{\epsilon^2}\right)$ | relative | log, hinge ReLU | Def. 1 | $\ell_1$ Lewis | Cors. 6, 9 |
| $\tilde{O}\left(\frac{d^3 \cdot \mu(X)^3}{\epsilon^4}\right)$ | relative | log | Def. 1 | sqrt lev. scores | [MSSW18] |
| $\tilde{O}\left(\frac{\sqrt{n} \cdot d^{3/2} \cdot \mu(X)}{\epsilon^2}\right)$ | relative | log | Def. 1 | sqrt lev. scores | [MSSW18] |
| $\tilde{O}\left(\frac{n^{1-\kappa}d}{\epsilon^2}\right)$ | relative | log, hinge | $\|x_i\|_2 \leq 1 \, \forall i$ regularization $n^\kappa \|\beta\|_1$, $n^\kappa \|\beta\|_2$, or $n^\kappa \|\beta\|_2^2$ | uniform | [CIM+19] |
| $O\left(\frac{\sqrt{d}}{\epsilon}\right)$ | additive $\epsilon n$ | log | $\|\beta\|_2, \|x_i\|_2 \leq 1 \, \forall i$ | deterministic | [KL19] |

Table 1: Comparison to prior work. $\tilde{O}(\cdot)$ hides logarithmic factors in the problem parameters. We note that the bounded norm assumption of [CIM+19] can be removed by simply scaling $X$, giving a dependence on the maximum row norm of $X$ in the sample complexity. The importance sampling distributions of our work and [MSSW18] are both in fact a mixture with uniform sampling. Our work and [KL19, CIM+19] generalize to broader classes of loss functions – for simplicity here we focus just on the important logistic loss, hinge loss, and ReLU.

**Experimental Evaluation.** In Sec 5, we compare our Lewis weight-based method to the square root of leverage score method of [MSSW18], uniform sampling as studied in [CIM$^+$19], and an oblivious sketching algorithm of [MOW21]. We study performance in minimizing both the log and hinge losses, with and without regularization. We observe that our method typically far outperforms uniform sampling, even in some cases when regularization is used. It performs comparably to the method of [MSSW18], seeming to outperform when the $\mu_y(X)$ complexity parameter is large.

## 1.2 Related Work

Our work is closely related to [MSSW18], which introduces the $\mu_y(X)$ complexity measure. They give relative error coresets with worse polynomial dependences on the parameters through a mixture of uniform sampling and sampling by the *squareroots of the leverage scores*. This approach has the same intuition as ours – the squareroot leverage score sampling preserves the 'linear part' of the hinge-like loss function and the uniform sampling preserves the asymptoting piece. However, like many other works on coresets for logistic regression and other problems [HCB16, TF18, CIM$^+$19, TMF20, TBFR21] the analysis of Munteanu et al. centers on the *sensitivity framework*. At best, this framework can achieve $\Omega(d^2)$ sample complexity – one $d$ factor comes from the total sensitivity of the problem, and the other from a VC dimension bound on the set of linear classifiers. To the best of our knowledge, our work is the first that avoids this sensitivity framework – Lewis weight sampling results are based on $\ell_1$ matrix concentration result and give optimal linear dependence on the dimension $d$.

**Regularized Classification Losses.** Rather than using the $\mu_y(X)$ parameterization of Def. 1, several other works [TF18, CIM$^+$19] achieve relative error coresets for the log and hinge losses by assuming that the loss function is regularized by $n^\kappa \cdot R(\beta)$, where $\kappa > 0$ is some parameter and $R(\beta)$ is some norm – e.g., $\|\beta\|_1$, $\|\beta\|_2$, or in the important case of soft-margin SVM, $\|\beta\|_2^2$.

Curtin et al. show that simple uniform sampling gives a relative error coreset with $\tilde{O}\left(n^{1-\kappa}d/\epsilon^2\right)$ points in this setting [CIM$^+$19]. They also show that no coreset with $o(n^{(1-\kappa)/5})$ points exists. In Appendix B, we tighten this lower bound, showing via a reduction to the INDEX problem in communication complexity that the $\tilde{O}(n^{1-\kappa})$ bound achieved by uniform sampling is in fact optimal.

Our theoretical results are incomparable to those of [CIM$^+$19]. However, empirically, Lewis weight sampling often far outperforms uniform sampling – see Sec. 5. Note that our results directly apply in the regularized setting – our relative error can only improve. However, our theoretical bounds do not actually improve with regularization, still depending on $\mu_y(X)$, which is avoided by [CIM$^+$19].

**Other Related Work.** Less directly, our work is connected to sampling and sketching algorithms for linear regression under different loss functions, often using variants of the leverage scores or Lewis weights [DDH$^+$09, CW14, ALS$^+$18, CWW19, CD21]. It is also related to work on sketching methods that preserve the norms of vectors under nonlinear transformations, like the ReLU, often with applications to coresets or compressed sensing for neural networks [BJPD17, BOB$^+$20, GM21]. Another line of related work includes coresets for standard optimization algorithms such as Frank-Wolfe algorithm [Cla10], Gilbert's algorithm [CHW12] and Stochastic Gradient Descent [HKS11].

## 2 Preliminaries

**Notation.** Throughout, for $f : \mathbb{R} \to \mathbb{R}$ and a vector $y \in \mathbb{R}^n$, we let $f(y) \in \mathbb{R}^n$ denote the entrywise application of $f$ to $y$. For a vector $y \in \mathbb{R}^n$ we let $y_i$ denote it's $i^{th}$ entry. So $f(y)_i = f(y_i)$.

For data matrix $X \in \mathbb{R}^{n \times d}$ with rows $x_1, \ldots, x_n \in \mathbb{R}^d$ and label vector $y \in \{-1, 1\}^n$ we consider classification loss functions of the form $L(\beta) = \sum_{i=1}^n f(\langle x_i, \beta \rangle \cdot y_i) = \sum_{i=1}^n f(D_y X \beta)_i$, where $D_y \in \mathbb{R}^{n \times n}$ is the diagonal matrix with $y$ on its diagonal. For simplicity, we write $X$ instead of $D_y X$ throughout, since we can think of the labels as just being incorporated into $X$ by flipping the signs of its rows. Similarly, we write the complexity parameter of Def. 1 as $\mu(X) = \sup_{\beta \neq 0} \frac{\|(X\beta)^+\|_1}{\|(X\beta)^-\|_1}$.

Throughout we will call $f(z) = \ln(1 + e^z)$ the *logistic loss* and $f(z) = \max(0, 1 + z)$ the *hinge loss*. Note that these functions have the sign of $z$ flipped from the typical convention. This is just notational – we can always negate $X$ or $\beta$ and have an identical loss function. We use these versions as they are both $\ell_\infty$ close to the ReLU function, a fact that we will leverage in our analysis.

**Basic sampling results.** Our coreset construction is based on sampling with the $\ell_1$ Lewis weights. We define these weights and state fundamental results on Lewis weight sampling and below.

**Definition 2** ($\ell_1$ Lewis Weights [CP15]). *For any $X \in \mathbb{R}^{n \times d}$ the $\ell_1$ Lewis weights are the unique values $\tau_1(X), \ldots, \tau_n(X)$ such that, letting $W \in \mathbb{R}^{n \times n}$ be the diagonal matrix with $1/\tau_1(X), \ldots, 1/\tau_n(X)$ as its diagonal, for all $i$,*

$$\tau_i(X)^2 = x_i^T (X^T W X)^+ x_i,$$

*where for any matrix $M$, $M^+$ is the pseudoinverse. $M^+ = M^{-1}$ when $M$ square and full-rank.*

**Theorem 3** ($\ell_1$ Lewis Weight Sampling). *Consider any $X \in \mathbb{R}^{n \times d}$, and set of sampling values $p_i$ with $\sum_{i=1}^n p_i = m$ and $p_i \geq \frac{c \cdot \tau_i(X) \log(m/\delta)}{\epsilon^2}$ for all $i$, where $c$ is a universal constant. If we generate a matrix $S \in \mathbb{R}^{m \times n}$ with each row chosen independently as the $i^{th}$ standard basis vector times $1/p_i$ with probability $p_i/m$ then there exists an $\ell > 1$ such that if $\sigma \in \{-1, 1\}^m$ is chosen with independent Rademacher entries*

$$\mathop{\mathbb{E}}_{S,\sigma} \left[ \sup_{\beta : \|X\beta\|_1 = 1} \left| \sum_{i=1}^m \sigma_i [SX\beta]_i \right|^\ell \right] \leq \epsilon^\ell \cdot \delta.$$

*In particular, if each $p_i$ is a scaling of a constant factor approximation to the Lewis weight $\tau_i(X)$ (which means that $\forall i, \ \tilde{c}\tau_i(X) \leq p_i \leq c\tau_i(X)$ for some constants $c, \tilde{c}$), $S$ has $m = O\left(\frac{d \log(d/\delta\epsilon)}{\epsilon^2}\right)$ rows.*

Theorem 3 is implicit in [CP15], following from the proof of Lemma 7.4, which shows a high probability bound on $|\|SX\beta\|_1 - 1|$ via the moment bound stated above. This moment bound is proven on page 29 of the arXiv version. The claim on the bound on $m$ can be derived as follows. We have $\sum_{i=1}^n \tau_i(X) = d$, which gives that $m = \sum p_i \leq cd \log(m/\delta)/\epsilon^2$ for some constant $c$. If we set $m = O(d \log(d/(\delta\epsilon))/\epsilon^2$ we have $\log(m/\delta) = O(\log(d/(\delta\epsilon))$ which gives $m = cd \log(m/\delta)/\epsilon^2$ as needed.

We will translate the above moment bound to give approximate bounds for classification loss functions like the ReLU, logistic loss, and hinge loss, using the following standard result on Rademacher complexities:

**Theorem 4** (Ledoux-Talagrand contraction, c.f. [Duc]). *Consider $V \subseteq \mathbb{R}^m$, along with $L$-Lipschitz functions $f_i : \mathbb{R} \to \mathbb{R}$ with $f_i(0) = 0$. Then for any $\ell > 1$, if $\sigma \in \{-1, 1\}^m$ is chosen with independent Rademacher entries,*

$$\mathop{\mathbb{E}}_\sigma \left[ \sup_{v \in V} \left| \sum_{i=1}^m \sigma_i f_i(v_i) \right|^\ell \right] \leq (2L)^\ell \cdot \mathop{\mathbb{E}}_\sigma \left[ \sup_{v \in V} \left| \sum_{i=1}^m \sigma_i v_i \right|^\ell \right].$$

# 3 Warm Up: Coresets for ReLU Regression

We start by showing that $\ell_1$ Lewis weight sampling yields a $(1 + \epsilon)$-relative error coreset for ReLU regression, under the complexity assumption of Def. 1. Our proofs for log loss, hinge loss, and other hinge-like loss functions will follow a similar structure, with some added complexities.

We first show that Lewis weight sampling gives a coreset with additive error $\epsilon\|X\|_1$. By setting $\epsilon' = \epsilon/\mu(X)$, we then easily obtain a relative error coreset under the assumption of Def. 1.

**Theorem 5** (ReLU Regression – Additive Error Coreset). *Consider $X \in \mathbb{R}^{n \times d}$ and let $\mathrm{ReLU}(z) = \max(0, z)$ for all $z \in \mathbb{R}$. For a set of sampling values $p_i$ with $\sum_{i=1}^n p_i = m$ and $p_i \geq \frac{c \cdot \tau_i(X) \log(m/\delta)}{\epsilon^2}$ for all $i$, where $c$ is a universal constant, if we generate $S \in \mathbb{R}^{m \times n}$ with each row chosen independently as the $i^{th}$ standard basis vector times $1/p_i$ with probability $p_i/m$ then with probability at least $1 - \delta$, for all $\beta \in \mathbb{R}^d$,*

$$\left| \sum_{i=1}^m [S\,\mathrm{ReLU}(X\beta)]_i - \sum_{i=1}^n \mathrm{ReLU}(X\beta)_i \right| \leq \epsilon\|X\beta\|_1.$$

*If each $p_i$ is a scaling of a constant factor approximation to the Lewis weight $\tau_i(X)$, $S$ has $m = O\left(\frac{d \log(d/(\delta\epsilon))}{\epsilon^2}\right)$ rows.*

**Corollary 6** (ReLU Regression – Relative Error Coreset). *Consider the setting of Thm. 5, where* $\sum_{i=1}^{n} p_i = m$ *and* $p_i \geq \frac{c \cdot \tau_i(X) \log(m/\delta) \cdot \mu(X)^2}{\epsilon^2}$ *for all $i$. With probability at least $1 - \delta$, $\forall \beta \in \mathbb{R}^d$,* $|\sum_{i=1}^{m} [S \operatorname{ReLU}(X\beta)]_i - \sum_{i=1}^{n} \operatorname{ReLU}(X\beta)_i| \leq \epsilon \cdot \sum_{i=1}^{n} \operatorname{ReLU}(X\beta)_i$. *If each $p_i$ is a scaling of a constant factor approximation to the Lewis weight $\tau_i(X)$, $S$ has $m = O\left(\frac{d \log(d/(\delta\epsilon)) \cdot \mu(X)^2}{\epsilon^2}\right)$ rows.*

*Proof of Corollary 6.* We have

$$\sum_{i=1}^{n} \operatorname{ReLU}(X\beta)_i = \sum_{i:[X\beta]_i \geq 0} [X\beta]_i = \|(X\beta)^+\|_1, \tag{2}$$

Additionally, since by definition $\mu(X) = \sup_{\beta \neq 0} \frac{\|(X\beta)^+\|_1}{\|(X\beta)^-\|_1} = \sup_{\beta \neq 0} \frac{\|(X\beta)^-\|_1}{\|(X\beta)^+\|_1}$,

$$\frac{\|X\beta\|_1}{\|(X\beta)^+\|_1} = 1 + \frac{\|(X\beta)^-\|_1}{\|(X\beta)^+\|_1} \leq 1 + \mu(X). \tag{3}$$

Combining (2) with (3) gives that $\sum_{i=1}^{n} \operatorname{ReLU}(X\beta)_i \geq \frac{1}{1+\mu(X)} \cdot \|X\beta\|_1$, which then completes the corollary after applying Theorem 5 with $\epsilon' = \frac{\epsilon}{1+\mu(X)}$. $\qquad\square$

*Proof of Theorem 5.* We prove the theorem restricted to $\beta$ such that $\|X\beta\|_1 = 1$. Since the ReLU function is linear in that $\operatorname{ReLU}(cz) = c \cdot \operatorname{ReLU}(z)$, this yields the complete theorem via scaling. It suffices to prove that there exists some $\ell > 0$ such that

$$B \stackrel{\text{def}}{=} \mathbb{E}_{S} \left[ \sup_{\beta : \|X\beta\|_1 = 1} \left| \sum_{i=1}^{m} [S \operatorname{ReLU}(X\beta)]_i - \sum_{i=1}^{n} \operatorname{ReLU}(X\beta)_i \right|^{\ell} \right] \leq \epsilon^{\ell} \cdot \delta.$$

The theorem then follows via Markov's inequality and the monotonicity of $z^{\ell}$ for $z, \ell \geq 0$. Via a standard symmetrization argument (c.f. the Proof of Theorem 7.4 in [CP15]) we have

$$B \leq 2^{\ell} \cdot \mathbb{E}_{S,\sigma} \left[ \sup_{\beta : \|X\beta\|_1 = 1} \left| \sum_{i=1}^{m} \sigma_i [S \operatorname{ReLU}(X\beta)]_i \right|^{\ell} \right],$$

where $\sigma \in \{-1, 1\}^m$ has independent Rademacher random entries. We can then apply, for each fixed value of $S$ the Ledoux-Talagrand contraction theorem (Theorem 4) with $V = \{SX\beta : \|X\beta\|_1 = 1\}$ and $f_i(z) = \operatorname{ReLU}(z)$ for all $i$. $f_i(z)$ is 1-Lipschitz with $f(0) = 0$. This gives

$$B \leq 4^{\ell} \cdot \mathbb{E}_{S,\sigma} \left[ \sup_{\beta : \|X\beta\|_1 = 1} \left| \sum_{i=1}^{m} \sigma_i [SX\beta]_i \right|^{\ell} \right] \leq (4\epsilon)^{\ell} \cdot \delta$$

for some $\ell > 1$ by Theorem 3. This completes the theorem after adjusting $\epsilon$ by a factor of 4, which only affects the sample complexity by a constant factor. $\qquad\square$

## 4  Extension to the Hinge Like Loss Functions

We next extend Theorem 5 to a family of 'nice hinge functions' which includes the hinge loss $f(z) = \max(0, 1 + z)$ and the log loss $f(z) = \ln(1 + e^z)$. These functions present two additional challenges: 1) they are generally not linear in that $f(c \cdot z) \neq c \cdot f(z)$, an assumption which is used in the proof of Theorem 5 to restrict to considering $\beta$ with $\|X\beta\|_1 = 1$ and 2) they are not contractions with $f(0) = 0$, a property which was used to apply the Ledoux-Talagrand contraction theorem.

**Definition 7** (Nice Hinge Function). *We call $f : \mathbb{R} \to \mathbb{R}^+$ an $(L, a_1, a_2)$-nice hinge function if for fixed constants $L, a_1$ and $a_2$,*

    *(1) $f$ is $L$-Lipschitz      (2) $|f(z) - \operatorname{ReLU}(z)| \leq a_1$ for all $z$      (3) $f(z) \geq a_2$ for all $z \geq 0$.*

We start with an additive error coreset result for nice hinge functions. We then show that under the additional assumption of $a_2 > 0$, the additive error achieved is small compared to $\sum_{i=1}^{n} f(X\beta)_i$, yielding a relative error coreset. This gives our main results for both the hinge loss and log loss, which are $(1, 1, 1)$-nice and $(1, \ln 2, \ln 2)$-nice hinge functions respectively.

**Theorem 8** (Nice Hinge Function – Additive Error Coreset). *Consider $X \in \mathbb{R}^{n \times d}$ and let $f : \mathbb{R} \to \mathbb{R}^+$ be an $(L, a_1, a_2)$-nice hinge function (Def. 7). For a set of sampling values $p_i$ with $\sum_{i=1}^n p_i = m$ and $p_i \geq \frac{C \max(\tau_i(X), 1/n)}{\epsilon^2}$ for all $i$, where $C = c \cdot \max(1, L, a_1)^2 \cdot \log\left(\frac{\log(n \max(1, L, a_1)/\epsilon) m}{\delta}\right)$ and $c$ is a fixed constant, if we generate $S \in \mathbb{R}^{m \times n}$ with each row chosen independently as the $i^{th}$ standard basis vector times $1/p_i$ with probability $p_i/m$, then with probability at least $1 - \delta$, $\forall \beta \in \mathbb{R}^d$,*

$$\left| \sum_{i=1}^m [Sf(X\beta)]_i - \sum_{i=1}^n f(X\beta)_i \right| \leq \epsilon \cdot (\|X\beta\|_1 + n).$$

Observe that for a fixed function $f$, $L, a_1$ are constant and so, if each $p_i$ is a scaling of a constant factor approximation to $\max(\tau_i(X), 1/n)$, $S$ has $m = O\left(\frac{d \log(\log(n/\epsilon) d/(\delta \epsilon))}{\epsilon^2}\right) = \tilde{O}\left(\frac{d}{\epsilon^2}\right)$ rows.

*Proof.* Let $J = c_1 \log(\frac{n \max(1, L, a_1)}{\epsilon})$ for some constant $c_1$. We will show that for each integer $j \in [-J, J]$, with probability at least $1 - \frac{\delta}{2J}$,

$$\sup_{\beta : \|X\beta\|_1 \in [2^j, 2^{j+1}]} \left| \sum_{i=1}^m [Sf(X\beta)]_i - \sum_{i=1}^n f([X\beta]_i) \right| \leq \epsilon \cdot 2^j + \epsilon \cdot n. \tag{4}$$

Via a union bound this gives the theorem for all $\beta$ with $\|X\beta\|_1 \in [2^{-J}, 2^J]$. We then just need to handle the case of $X\beta$ with norm outside this range – i.e. when $\|X\beta\|_1$ is polynomially small or polynomially large in $n$ and the other problem parameters. We will take a union bound over the failure probabilities for these cases, and after adjusting $\delta$ by a constant, have the complete theorem. We make the argument for $\|X\beta\|_1$ outside $[2^{-J}, 2^J]$ first.

**Small Norm.** For $\beta$ with $\|X\beta\|_1 \leq 2^{-J}$, $\|X\beta\|_\infty \leq 2^{-J} \leq \frac{\epsilon}{L}$. Thus, $f(X\beta)_i \in [f(0) - \epsilon, f(0) + \epsilon]$ for all $i$. Thus by triangle inequality, and the fact that $f(0) \leq \text{ReLU}(0) + a_1 = a_1$:

$$\sup_{\beta : \|X\beta\|_1 \leq 2^{-J}} \left| \sum_{i=1}^m [Sf(X\beta)]_i - \sum_{i=1}^n f(X\beta)_i \right| \leq a_1 \cdot \left| \sum_{i=1}^m 1/p_{j_i} - n \right| + \epsilon \cdot \left( \sum_{i=1}^m 1/p_{j_i} + n \right), \tag{5}$$

where $1/p_{j_i}$ is value of the single nonzero entry in the $i^{th}$ row of $S$, which samples index $j_i$ from $f(X\beta)$. Let $Z_1, \ldots, Z_m$ be i.i.d., each taking value $1/p_j$ with probability $\frac{p_j}{m}$ for all $j \in [n]$. Then

$$\left| \sum_{i=1}^m 1/p_{j_i} - n \right| = \left| \sum_{i=1}^m (Z_i - \mathbb{E} Z_i) \right|.$$

For all $j \in [n]$ we have $p_j \geq \frac{C \cdot \max(\tau_j(X), 1/n)}{\epsilon^2} \geq \frac{c \cdot \max(1, a_1)^2 \cdot \log(J/\delta)}{n \epsilon^2}$, so applying a Bernstein bound, if the constant $c$ is chosen large enough we have:

$$\mathbb{P}\left[ \left| \sum_{i=1}^m 1/p_{j_i} - n \right| \geq \frac{\epsilon n}{\max(1, a_1)} \right] \leq 2 \exp\left( -\frac{\epsilon^2 n^2 / (2 \max(1, a_1)^2)}{\frac{n^2 \epsilon^2}{c \log(J/\delta) \cdot \max(1, a_1)^2} + \frac{n^2 \epsilon^3}{c \log(J/\delta) \cdot \max(1, a_1)^3}} \right) \leq \frac{\delta}{4J}. \tag{6}$$

Combining (6) with (5), with probability at least $1 - \delta$, we have

$$\sup_{\beta : \|X\beta\|_1 \leq 2^{-J}} \left| \sum_{i=1}^m [Sf(X\beta)]_i - \sum_{i=1}^n f(X\beta)_i \right| \leq \frac{a_1 \cdot \epsilon n}{\max(1, a_1)} + \epsilon \cdot \left( 2 + \frac{\epsilon}{\max(1, a_1)} \right) n \leq 4\epsilon \cdot n.$$

Adjusting constants on $\epsilon$, this gives the theorem for $\beta$ with $\|X\beta\|_1 \leq 2^{-J}$.

**Large Norm.** We next consider $\beta$ with $\|X\beta\|_1 \geq 2^J$. Since by assumption $|f(z) - \text{ReLU}(z)| \leq a_1$ for all $z \in \mathbb{R}$, we can apply triangle inequality to give for any $\beta \in \mathbb{R}^d$,

$$\left| \sum_{i=1}^m [Sf(X\beta)]_i - \sum_{i=1}^n f(X\beta)_i \right| \leq \left| \sum_{i=1}^m [S\,\text{ReLU}(X\beta)]_i - \sum_{i=1}^n \text{ReLU}(X\beta)_i \right| + a_1 \cdot \left( \sum_{i=1}^m 1/p_{j_i} + n \right).$$

Applying Theorem 5 and the bound on $\sum_{i=1}^m 1/p_{j_i}$ given in (6), we thus have, with probability at least $1 - 2\delta$, for all $\beta$ with $\|X\beta\|_1 \geq 2^J$,

$$\left|\sum_{i=1}^m [Sf(X\beta)]_i - \sum_{i=1}^n f(X\beta)_i\right| \leq \frac{\epsilon}{2}\|X\beta\|_1 + 3a_1 n \leq \epsilon\|X\beta\|_1,$$

where the final bound uses that $\|X\beta\|_1 \geq 2^J \geq \left(\frac{n\max(1,a_1)}{\epsilon}\right)^{c_1}$ for a large enough constant $c_1$. This gives the theorem for $\beta$ with $\|X\beta\|_1 \geq 2^J$.

**Bounded Norm.** We now return to proving that (4) holds for any $j \in [-J, J]$ with probability at least $1 - \frac{\delta}{2J}$. Let $\bar{f}(z) = f(z) - f(0)$. Then for any $\beta \in \mathbb{R}^d$ we have:

$$\left|\sum_{i=1}^m [Sf(X\beta)]_i - \sum_{i=1}^n f([X\beta]_i)\right| \leq \left|\sum_{i=1}^m [S\bar{f}(X\beta)]_i - \sum_{i=1}^n \bar{f}([X\beta]_i)\right| + f(0) \cdot \left|\sum_{i=1}^m 1/p_{j_i} - n\right|.$$

We again apply the bound on $\sum_{i=1}^m 1/p_{j_i}$ given in (6) and the fact that $f(0) \leq a_1$. This gives that with probability at least $1 - \frac{\delta}{4J}$, for all $\beta \in \mathbb{R}^d$,

$$\left|\sum_{i=1}^m [Sf(X\beta)]_i - \sum_{i=1}^n f([X\beta]_i)\right| \leq \left|\sum_{i=1}^m [S\bar{f}(X\beta)]_i - \sum_{i=1}^n \bar{f}([X\beta]_i)\right| + f(0) \cdot \frac{\epsilon n}{\max(1, a_1)}$$

$$\leq \left|\sum_{i=1}^m [S\bar{f}(Xx)]_i - \sum_{i=1}^n \bar{f}([Xx]_i)\right| + \epsilon \cdot n. \tag{7}$$

Now, for $\ell \geq 0$, by a standard symmetrization argument (c.f. the proof of Thm. 7.4 in [CP15]),

$$B \stackrel{\text{def}}{=} \mathbb{E}_S\left[\sup_{\beta: \|X\beta\|_1 \in [2^j, 2^{j+1}]}\left|\sum_{i=1}^m [S\bar{f}(X\beta)]_i - \sum_{i=1}^n \bar{f}(X\beta)_i\right|^\ell\right] \leq 2^\ell \mathbb{E}_{S,\sigma}\left[\sup_{\beta: \|X\beta\|_1 \in [2^j, 2^{j+1}]}\left|\sum_{i=1}^n \sigma_i [S\bar{f}(X\beta)]_i\right|^\ell\right],$$

where $\sigma \in \{-1, 1\}^m$ has independent Rademacher random entries. We can then apply, for each fixed value of $S$ the Ledoux-Talagrand contraction theorem (Thm. 4) with $V = \{SX\beta : \|X\beta\|_1 \in [2^j, 2^{j+1}]\}$ and $f_i(z) = 1/p_{j_i} \cdot \bar{f}(p_{j_i} \cdot z)$. Note that $f_i(0) = 0$ since $\bar{f}(0) = 0$. Additionally, $f_i$ is $L$-Lipschitz since by assumption $f(z)$ is $L$-Lipschitz so $\bar{f}(p_{j_i} \cdot z)$ is $(p_{j_i} \cdot L)$-Lipschitz. We have,

$$f_i([SX\beta]_i) = 1/p_{j_i} \cdot \bar{f}(p_{j_i} \cdot 1/p_{j_i}[X\beta]_{j_i}) = [S\bar{f}(X\beta)]_i.$$

So, applying Theorem 3, for some $\ell > 1$ we have, since $p_i \geq \frac{C\tau_i(X)}{\epsilon^2}$ for $C = c\max(1, L, a_1)^2 \cdot \log(\log(n\max(1, L, a_1)/\epsilon)m/\delta) = \Omega(\max(1, L^2) \cdot \log(Jm/\delta))$,

$$B \leq (4L)^\ell \cdot \mathbb{E}_{S,\sigma}\left[\sup_{\beta: \|X\beta\|_1 \in [2^j, 2^{j+1}]}\left|\sum_{i=1}^n \sigma_i [SX\beta]_i\right|^\ell\right] \leq (4L)^\ell \cdot (\epsilon/L)^\ell \cdot \frac{\delta}{4J} \cdot (2^{j+1})^\ell.$$

Adjusting $\epsilon$ by a constant, this gives via Markov's inequality that with probability at least $1 - \frac{\delta}{4J}$,

$$\sup_{\beta: \|X\beta\|_1 \in [2^j, 2^{j+1}]}\left|\sum_{i=1}^m [S\bar{f}(X\beta)]_i - \sum_{i=1}^n \bar{f}(X\beta)_i\right| \leq \epsilon \cdot 2^j. \tag{8}$$

In combination with (7), we then have that probability at least $1 - \frac{\delta}{2J}$,

$$\sup_{\beta: \|X\beta\|_1 \in [2^j, 2^{j+1}]}\left|\sum_{i=1}^m [Sf(X\beta)]_i - \sum_{i=1}^n f([X\beta]_i)\right| \leq \epsilon \cdot 2^j + \epsilon \cdot n.$$

This gives (4) and completes the theorem. $\qquad\square$

## 4.1 Relative Error Coresets

Our relative error coreset result for nice hinge functions follows as a simple corollary of Thm. 8. The proof is analogous to the proof of Cor. 6, and given in the appendix.

**Corollary 9** (Nice Hinge Function – Relative Error Coreset). *Consider the setting of Thm. 8 under the additional assumption that $a_2 > 0$. If $\sum_{i=1}^{n} p_i = m$ and $p_i \geq \frac{C \max(\tau_i(X), 1/n) \cdot \mu(X)^2}{\epsilon^2}$ for all $i$, where $C = c \cdot \max(1, L, a_1, 1/a_2)^{10} \cdot \log \left( \frac{\log(n \max(1, L, a_1, 1/a_2) \cdot \mu(X)/\epsilon)m}{\delta} \right)$ and $c$ is a fixed constant, with probability $\geq 1 - \delta$, for all $\beta \in \mathbb{R}^d$, $|\sum_{i=1}^{m} [Sf(X\beta)]_i - \sum_{i=1}^{n} f(X\beta)_i| \leq \epsilon \cdot \sum_{i=1}^{n} f(X\beta)_i$.*

For fixed $f(\cdot)$, $L, a_1, a_2$ are constant and so, if each $p_i$ is a scaling of a constant factor approximation to $\max(\tau_i(X), 1/n)$, $S$ has $m = O\left( \frac{d\mu(X)^2 \log(\log(n\mu(X)/\epsilon)d\mu(X)/(\delta\epsilon))}{\epsilon^2} \right) = \tilde{O}\left( \frac{d\mu(X)^2}{\epsilon^2} \right)$ rows. This gives our main result for the hinge and log losses, which are $(1, 1, 1)$ and $(1, \ln 2, \ln 2)$-nice.

# 5 Empirical Evaluation

We now compare our method (`lewis`), square root of leverage score method (`l2s`) of [MSSW18], uniform sampling (`uniform`), and an oblivious sketching algorithm (`sketch`) of [MOW21]. Our evaluation uses the codebase of [MSSW18], which was generously shared with us.

**Implementation.** Lewis weights are computed via an iterative algorithm given in [CP15], which involves computing leverage scores of a reweighted input matrix in each iteration. We typically don't need many iterations to reach convergence – for all datasets we used 20 iterations and observed relative difference between successive iterations around $10^{-6}$. Leverage scores are also needed by the `l2s` routine, and are computed via the numpy *qr* factorization routine when possible. One dataset (COVERTYPE) involves an almost singular matrix, we resorted to the *pinv* routine in numpy.

We note that [MSSW18] used a fast random sketching-based QR decomposition to compute the leverage scores – this can also be applied to Lewis weight computation. The number of iterations of the Lewis weight algorithm can also be reduced – it seems that roughly 5 iterations are sufficient for practical purposes. Lewis weight computation will then take about 5 times as much time as `l2s` weight computation.

**Datasets.** We use the same three datasets as in [MSSW18]. The WEBB SPAM[1] data consists of 350,000 unigrams with 127 features from web pages with 61% positive labels. The task is is to classify as spam or not. The other two datasets are loaded from scikit learn library[2]. COVERTYPE consists of 581,012 cartographic observations of different forests with 54 features and 49% positive labels. The task is to predict the type of tree. KDD CUP '99 has 494,021 points with 41 features and 20% positive labels. The task is to detect network intrusions.

**Loss functions.** We evaluate the algorithms on two loss functions: 1) logistic loss $f(z) = \ln(1 + e^z)$ and 2) hinge loss $f(z) = \max(0, 1+z)$. As before, we use $z = \langle \beta, x \rangle \cdot y$. Note that [MSSW18] gives guarantees only for logistic loss for `l2s`. We also evaluate the above two losses with regularization term $0.5\|\beta\|_2^2$. We evaluate `sketch` only for logistic loss without any regularization – which is what is was designed for. Though it sometime preforms reasonably in other cases, it can have very high variance or high error for certain combinations of loss functions and datasets.

**Evaluation.** Our evaluation follows that of [MSSW18]. Let $\tilde{\beta}$ be the parameter vector minimizing the sum of the loss function on the coreset and $\beta^*$ be the true minimizer. We report the relative loss $\frac{|L(\beta^*) - L(\tilde{\beta})|}{L(\beta^*)}$, where $L(\beta) = \sum_i^n f(\langle x_i, \beta \rangle \cdot y_i)$ is the sum of loss over all data points. Ideally, this ratio should be close to 0. In Figure 1, we plot the log relative loss as a function of coreset size.

We observe that Lewis weights sampling performs better than all other methods on KDD CUP '99 for both loss functions, with and without regularization. Our bounds for `lewis` give a better dependence on the complexity parameter $\mu_y(X)$ than the bounds [MSSW18] for `l2s`, and so this agrees with the fact that the value of $\mu_y(X)$ is high for KDD CUP '99. [MSSW18] estimated $\mu_y(X)$ values of WEBB SPAM, COVERTYPE and KDD CUP '99 to be 4.39, 1.86 and 35.18 respectively. For COVERTYPE and

---

[1]https://www.csie.ntu.edu.tw/ cjlin/libsvmtools/datasets/

[2]https://scikit-learn.org/

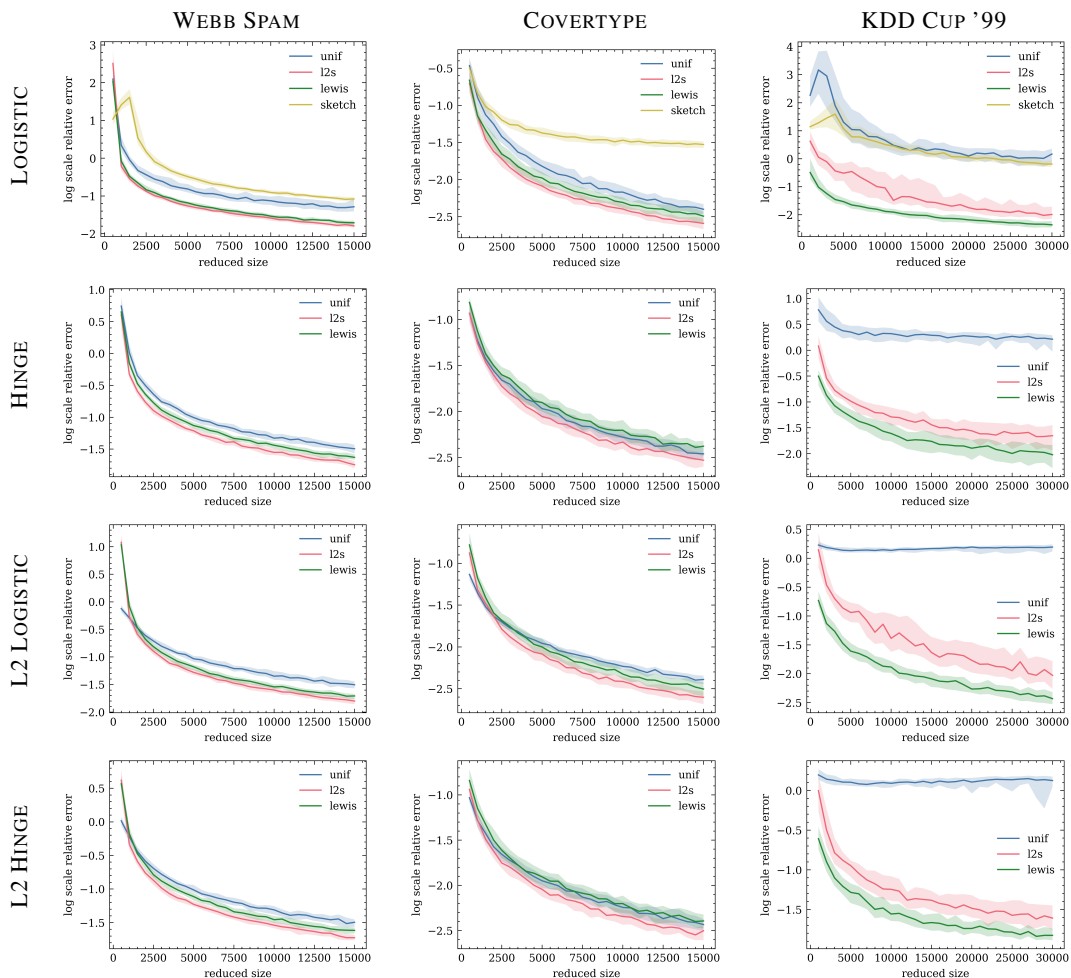

Figure 1: Each plot represents the performance of the studied methods with respect to one combination of dataset and loss function. The $x$-axis shows the coreset size, and the $y$-axis shows the log scale of the relative error. Each experiment is run 100 times. The solid line represents the median error. The lower and upper boundary lines represent the 25th and 75th percentiles respectively.

WEBB SPAM, the performance of `lewis` is comparable or a little worse than that of `l2s`. Furthermore, on these two datasets, with regularization, uniform sampling does relatively well for very small sample sizes, which agrees with the results of [CIM+19].

**Comparison of distributions.** To give a better intuition behind our results, we illustrate how different the Lewis weights are from the other sampling distributions on our three datasets in Fig. 3. Given two distributions $\bar{p} = (p_1, p_2, .., p_n)$ and $\bar{q} = (q_1, q_2, .., q_n)$, we plot the frequencies of $\{\max(p_i/q_i, q_i/p_i)\}_i$. We let $\bar{p}$ to be the uniform or `l2s` distributions and take $\bar{q}$ to be Lewis weights. We observe that the Lewis weights are far from uniform on all datasets, especially KDD CUP '99. This may explain why `lewis` performs so well on this dataset. `l2s` and `lewis` are much closer in general, explaining their relatively similar performance. Note that these score comparisons are based only on the data matrix $X$, and not the label vector $y$, which does not affect the leverage scores or Lewis weights. Thus, they only give a partial picture of the differences between methods. In particular, our theoretical bounds and the bounds for `l2s` in [MSSW18] both depend on $\mu_y(X)$, which depends on the label vector.

**Classification Error** Here, we include additional experimental results based on reviewer feedback. For various coreset sizes and datasets, we plot the $\log(1 - \text{auc})$ for each of the methods. *auc* stands

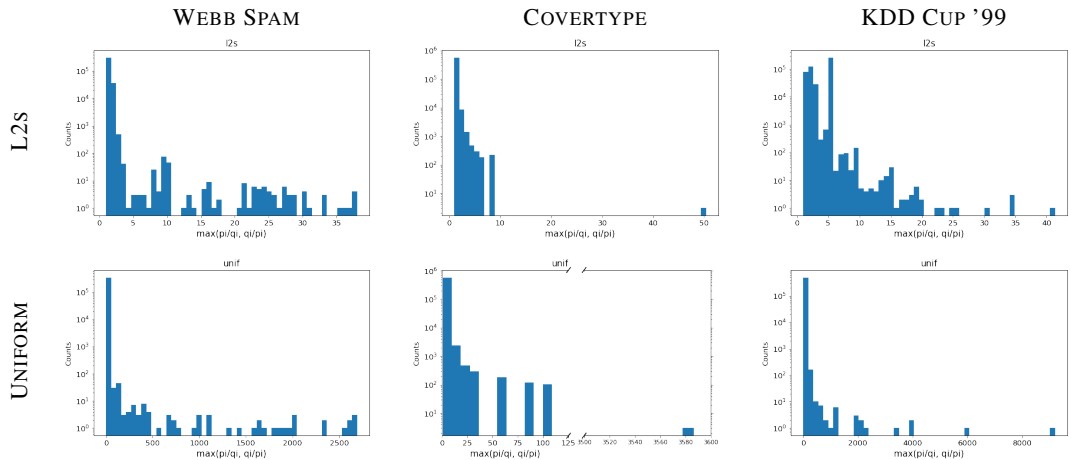

Figure 2: Comparison of sampling distributions.

for *Areas Under the Curve*. A perfect classifier has auc = 1 and all other classifiers have auc < 1. The lower the curve is in the plot, better it is as a classifier. Each plot is a result of 100 trials.

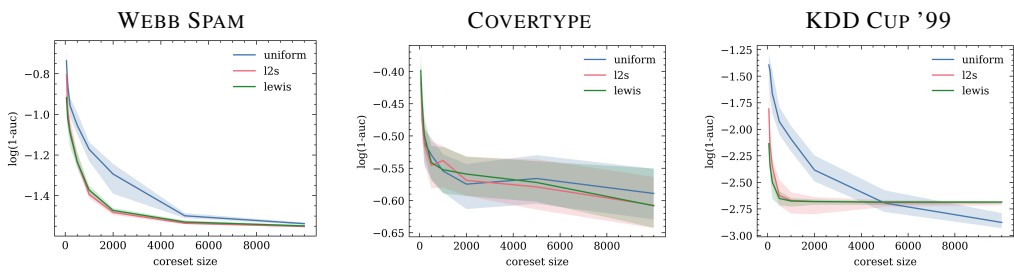

Figure 3: Comparison of classification accuracy.

## Acknowledgments and Disclosure of Funding

Cameron Musco was partially supported by an Adobe Research grant, along with NSF Grants 2046235 and 1763618.

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
