## A Omitted Proofs

**Corollary 9** (Nice Hinge Function – Relative Error Coreset). *Consider the setting of Thm. 8 under the additional assumption that $a_2 > 0$. If $\sum_{i=1}^n p_i = m$ and $p_i \geq \frac{C \max(\tau_i(X), 1/n) \cdot \mu(X)^2}{\epsilon^2}$ for all $i$, where $C = c \cdot \max(1, L, a_1, 1/a_2)^{10} \cdot \log\left(\frac{\log(n \max(1, L, a_1, 1/a_2) \cdot \mu(X)/\epsilon)m}{\delta}\right)$ and $c$ is a fixed constant, with probability at least $1 - \delta$, for all $\beta \in \mathbb{R}^d$,*

$$\left| \sum_{i=1}^m [Sf(X\beta)]_i - \sum_{i=1}^n f(X\beta)_i \right| \leq \epsilon \cdot \sum_{i=1}^n f(X\beta)_i.$$

*Proof.* By (3) proven in Corollary 6 and using the fact that $f$ is $(L, a_1, a_2)$-nice,

$$\sum_{i=1}^n f(X\beta)_i \geq \sum_{i:[X\beta]_i \in [0, 2a_1]} f(X\beta)_i + \sum_{i:[X\beta]_i \geq 2a_1} f(X\beta)_i$$

$$\geq \sum_{i:[X\beta]_i \in [0, 2a_1]} a_2 + \sum_{i:[X\beta]_i \geq 2a_1} \text{ReLU}(X\beta)_i - a_1$$

$$\geq \min\left(\frac{a_2}{2a_1}, \frac{1}{2}\right) \cdot \|(X\beta)^+\|_1$$

$$\geq \min\left(\frac{a_2}{2a_1}, \frac{1}{2}\right) \cdot \frac{\|X\beta\|_1}{\mu(X) + 1}. \tag{9}$$

Let $\gamma \overset{\text{def}}{=} \min\left(\frac{a_2}{2a_1}, \frac{1}{2}\right)$. Now we claim that $\sum_{i=1}^n f(X\beta)_i \geq \frac{na_2\gamma}{4\max(1,L)\cdot\mu(X)}$. If $\sum_{i=1}^n f(X\beta)_i \geq \frac{na_2}{4}$ then this holds immediately since $\mu(X) \geq 1$, $\max(1, L) \geq 1$ and $\gamma \leq 1$. Otherwise, assume that $\sum_{i=1}^n f(X\beta)_i \leq \frac{na_2}{4}$. Since $f(z) \geq a_2$ for all $z \geq 0$ and since $f$ is $L$-Lipschitz, $f(z) \geq \frac{a_2}{2}$ for all $z \geq -\frac{a_2}{2L}$. This implies that $X\beta$ has at most $\frac{na_2/4}{a_2/2} = \frac{n}{2}$ entries $\geq -\frac{a_2}{2L}$. Thus, $X\beta$ has at least $\frac{n}{2}$ entries $\leq -\frac{a_2}{2L}$ and so $\|(X\beta)^-\|_1 \geq \frac{na_2}{4L}$. Thus, by the definition of $\mu(X)$ along with (9),

$$\sum_{i=1}^n f(X\beta)_i \geq \gamma \cdot \|(X\beta)^+\|_1 \geq \frac{na_2\gamma}{4L \cdot \mu(X)} \geq \frac{na_2\gamma}{4\max(1,L)\cdot\mu(X)}. \tag{10}$$

Combining (9) with (10) gives that

$$\sum_{i=1}^n f(X\beta)_i \geq \frac{\gamma \cdot \|X\beta\|_1}{2\mu(X) + 2} + \frac{na_2\gamma}{8\max(1,L)\cdot\mu(X)} \geq (\|X\beta\|_1 + n) \cdot \frac{\gamma \cdot \min(1, a_2)}{8\max(1,L)\cdot\mu(X) + 2}.$$

This completes the corollary after applying Thm. 8 with

$$\epsilon' = \epsilon \cdot \frac{\gamma \cdot \min(1, a_2)}{8\max(1,L)\cdot\mu(X) + 2} \geq \frac{\epsilon}{8\max(1, L, a_1, 1/a_2)^4 \cdot \mu(X) + 2}.$$

$\square$

## B Lower Bounds for Regularized Classification

We now give a lower bound showing that the results of [CIM$^+$19] on coresets for regularized logistic and hinge loss regression (i.e., soft margin SVM) are essentially tight. Our bound tightens a lower bound given in [CIM$^+$19]. It shows that, in the natural setting where the regularization parameter is sublinear in the number of data points $n$, the coreset size must depend polynomially on $n$. This contrasts the setting where we assume that $\mu(X)$ from Def. 1 is bounded. In this case, as shown in Cor. 9, relative error coresets with size scaling just logarithmically in $n$ are achievable.

**Theorem 10** (Regularized Classification – Relative Error Lower Bound). *Let $X \in \mathbb{R}^{n \times d}$ have all row norms bounded by 1. Let $f$ be the hinge loss $f(z) = \max(0, 1 + z)$ or log loss $f(z) = \ln(1 + e^z)$ and for any $\kappa \in (0, 1)$ consider the regularized loss $L : \mathbb{R}^d \to \mathbb{R}^+$,*

$$L(\beta) = \sum_{i=1}^n f(X\beta)_i + n^\kappa \cdot R(\beta),$$

*where $\kappa \in (0,1)$. There is no $O(1)$ relative error coreset for $L(\beta)$ with $o\left(\frac{n^{1-\kappa}}{\log^c n}\right)$ points where $c = 4$ for $R(\beta) = \|\beta\|_2^2$, $c = 5/2$ for $R(\beta) = \|\beta\|_2$, and $c = 3$ for $R(\beta) = \|\beta\|_1$.*

Note that since this is a lower bound, the assumption that $X$ has bounded row norms only makes it stronger. This assumption is common in prior work.

*Proof.* We focus on the case when $f$ is the hinge loss for simplicity. An identical argument applies when $f$ is the log loss, with some adjustments of the constants. We also focus on the case when $R(\beta) = \|\beta\|_2^2$. Again, essentially an identical argument proves the claim when $R(\beta) = \|\beta\|_2$ or $R(\beta) = \|\beta\|_1$. We prove the lower bound via a reduction from the INDEX problem in communication complexity. Alice has a string $a \in \{0,1\}^n$ and Bob has an index $b \in \{1, \ldots, n\}$, and they wish to compute the bit $a(b)$. It is well known that the randomized 1-way communication complexity of this problem is $\Omega(n)$ [Rou15]. We will show that the existence of a relative error coreset for $L(x)$ with $o\left(\frac{n^{1-\kappa}}{\log^4 n}\right)$ points would contradict this lower bound, giving the result.

Assume without loss of generality that $n^{1-\kappa}$ is a power of two. Let $d = \log_2 n^{1-\kappa}$. Our reduction is to the INDEX problem with input size $n_0 = \frac{n^{1-\kappa}}{d(d+1)^2} = \Theta\left(\frac{n^{1-\kappa}}{\log^3 n}\right)$. Let Alice construct the matrix $X_0 \in \mathbb{R}^{n_0 \times (d+1)}$ which has the first $d$ entries of row $i$ equal to the binary representation of $i$ if $a(i) = 1$ and equal to 0 otherwise. In the binary representation, have 0 represented by $-1$ and 1 represented by 1. Let every row have $d$ in the last column. Finally, scale the matrix by a $\gamma = 1/\sqrt{d^2 + d}$ factor so each row has Euclidean norm exactly 1. Let $X \in \mathbb{R}^{n \times (d+1)}$ be equal to $n^\kappa \cdot d(d+1)^2$ copies of $X_0$ stacked on top of each other (assume without loss of generality that $n^\kappa \cdot d(d+1)^2$ is an integer).

Bob will let $\beta \in \mathbb{R}^{d+1}$ be the binary representation for $b$ (again written using $-1$s and 1s) with a $-1$ in the last entry. He will scale $\beta$ by a $1/\gamma$ factor so $\|\beta\|_2^2 = (d+1) \cdot (d^2 + d) = d(d+1)^2$. If $a(b) = 1$ we have:

$$L(\beta) = n^\kappa \cdot d(d+1)^2 \cdot \left(\sum_{j \neq b} h(X\beta)_j + h(X\beta)_b\right) + n^\kappa \|\beta\|_2^2$$

$$= n^\kappa \cdot d(d+1)^2 + n^\kappa \cdot d(d+1)^2 = 2n^\kappa \cdot d(d+1)^2, \tag{11}$$

where the second line holds since for $j \neq b$, $[X\beta]_j \leq d - 1 - d \leq -1$ and so $h(X\beta)_j = 0$. $[X\beta]_b = d - d = 0$ and so $h(X\beta)_b = 1$. Otherwise, by the same logic, if $a(b) = 0$ we have:

$$L(\beta) = n^\kappa \cdot d(d+1)^2 \cdot \left(\sum_{j \neq b} h(X\beta)_j + h(X\beta)_b\right) + n^\kappa \|\beta\|_2^2 = n^\kappa \cdot d(d+1)^2. \tag{12}$$

From (11) and (12), we can see that a coreset with relative error $\epsilon = 1/2$ can distinguish the two cases of $a(b) = 1$ and $a(b) = 0$. Assume that there is such a relative error coreset consisting of $m$ rows of $X$, along with $m$ corresponding weights $w_1, \ldots, w_m$. We can assume that all $w_j \leq n^{c_1}$ for some large constant $c_1$. If $a(i_j) = 1$ any $w_j$ larger than this would lead to the coreset cost being a large over estimate when $b = i_j$. If $a(i_j) = 0$, then scaling the $i_j^{th}$ row by any $w_j$ will have no effect since for all $\beta$ that Bob may generate, $h(X\beta)_{i_j} = 0$. So again, we can assume $w_j \leq n^{c_1}$.

Additionally, if we round each $w_j$ to the nearest integer multiple of $1/n^{c_1}$ we will not change the coreset cost by more than a $n/n^{c_1}$ factor in all our input cases, since we always have $h(X\beta)_i \in [0,1]$. Thus, Alice can represent each rounded $w_j$ using $\log n$ bits and send the full coreset and weights to Bob using $O(m \cdot (\log n + d)) = O(m \log n)$ bits of communication. Since Bob can then use this coreset to solve the INDEX with input size $n_0 = \Theta\left(\frac{n^{1-\kappa}}{\log^3 n}\right)$, we must have $m = \Omega\left(\frac{n^{1-\kappa}}{\log^4 n}\right)$, proving the theorem.

In the case that $R(\beta) = \|\beta\|_2$ we have $\|\beta\|_2 = d^{1/2}(d+1) = \Theta(d^{3/2})$ and so can set $n_0 = \Theta\left(\frac{n^{1-\kappa}}{\log^{3/2} n}\right)$ instead of $n_0 = \Theta\left(\frac{n^{1-\kappa}}{\log^3 n}\right)$, which gives the final lower bound of $\Omega\left(\frac{n}{\log^{5/2} n}\right)$. Similarly, for $R(\beta) = \|\beta\|_1$, we have $\|\beta\|_1 = d^{1/2}(d+1)^{3/2} = \Theta(d^2)$, yielding a final bound of $\Omega\left(\frac{n}{\log^3 n}\right)$. $\qquad\square$

Finally, we compare our lower bound with the the bound in [TBFR21]. We first note that the lower bound in the referenced paper is a lower bound on the sum of sensitivities, rather than directly on the coreset size, as we have given. We are not aware of a general result which lower bounds coreset size via the sum of sensitivities, although perhaps such a result could be shown, at least for reasonable classes of loss functions.

If we set $\lambda$ in [TBFR21] to $n^{-\kappa}$, then we are in the same setting as our lower bound, with regularization $n^{\kappa}\|\beta\|$. In this setting, assuming that $d < n^{1-\kappa}$, then the lower bound given in Lemma 1 of [TBFR21] is $O(n\lambda/d^2) = O(n^{1-\kappa}/d^2)$. This is loose by a $d^2$ factor, as compared to our tight lower bound.