# OpenReview forum: "Coresets for Classification – Simplified and Strengthened"
_NeurIPS.cc/2021/Conference — NeurIPS 2021 Poster_

### Official Review · Reviewer_YW1c · 2021-07-05

**Rating:** 7
**Confidence:** 4

**Summary:**

This paper deals with constructing coresets using $\ell_1$ Lewis weights. Such problems usually focused on $\ell_p$-regression problems but in this paper, a coreset construction scheme was applied to a fairly general class of functions, namely, "nice hinge loss" functions, including but not restricted to the RELU loss function, log hinge function, hinge function, etc.

As for applications, this paper generalizes existing work on logistic regression (without the regularization) using lewis weights. It also discusses a coreset construction for the hinge loss. The authors claim the first result of coreset construction concerning "nice hinge loss functions" that do not depend on the sensitivity framework. This enables avoiding the extra multiplicative $O(d^2)$ term in the sample complexity.

In general, I like this paper, however, I have some comments concerning the Related Work (see below).

**Limitations And Societal Impact:**

As stated in the paper, similar to [MSSW18], a valid assumption (that was used in this paper), is that the classification complexity is small, i.e., datasets with linear separability are avoided, since the classification complexity would then approaches infinity. In addition, the theoretical bounds obtained in the paper do not improve with regularization.

**Main Review:**

The paper suggests a coreset construction scheme for a family of functions, namely "nice hinge loss functions", using a none sensitivity framework, i.e., using Lewis weights, specifically, $\ell_1$ Lewis weights. This work builds upon several works and mainly improves upon and generalizes that of [MSSW18] to a larger family of functions.

In the paper, the sample complexity depends quadratically on the classification complexity, a notion that was first defined in [MSSW18], yielding a better and smaller bound than previously achieved. In addition, in the paper, a tighter lower bound on the sample complexity has been established with respect to the regularized problems such as the hinge loss and log hinge loss (variants of the SVM problem and logistic regression problem).

I believe the results are correct. I have looked into most of the proofs though not all, due to time constraints.

There are some typos which I have come upon:
- The equation below Line 119, change n to m in the sum term.
-  At the supplementary, Line 410, there is an addition $\leq$.
- At the supplementary, Line 455, $n_0$ instead of $n^0$.

I believe that the related work is not updated enough, as it lacks discussion of papers that admit coreset construction with respect to logistic regression and SVMs! To name some:
- On coreset for Support Vector Machines
- Coresets for Near-Convex Functions
- Coresets, sparse greedy approximation, and the Frank-Wolfe algorithm

Questions:
- How does your lower bound on the regularized version of the hinge loss (a variant of SVM) compare with the lower bound obtained at "On coreset for Support Vector Machines"?
- How does your upper bound on the sample complexity compare with that of "Coresets for Near-Convex Functions" for either logistic regression or SVM with respect to the regularized loss function?
- In some cases, the $\ell_2$-Lewis weights are 2-3 times better than your approach in terms of the approximation error for the sample size. Can you give a brief explanation of why this happens?
- When does your approach outperform that of [MSSW18] (practically speaking)?
- It would be good to see a comparison between your approach and sensitivity-based approaches.
- Can you plot the approximation error as a function of time?


**Time Spent Reviewing:**

6 hours

---

> ### Author Response · Authors · 2021-08-09
> **We compare our lower bound with that in "On coreset for Support Vector Machines"  and also respond to many other questions.**
>
> Thanks for the positive review, and for pointing out several typos, which we will correct. Thank you also for pointing out these related works, which we missed. We will definitely add a discussion of them.
>
> __"How does your lower bound...compare with the lower bound obtained at "On coreset for Support Vector Machines?"__
>
> This is indeed a closely related bound, which we will add a discussion of. We first note that the lower bound in the referenced paper is a lower bound on the sum of sensitivities, rather than directly on the coreset size, as we have given. We are not aware of a general result which lower bounds coreset size via the sum of sensitivities, although perhaps such a result could be shown, at least for reasonable classes of loss functions.
>
> If we set $\\lambda$ in the referenced paper to $n^{-\\kappa}$, then we are in the same setting as our lower bound, with regularization $n^{\\kappa} \\cdot \\| \\beta \\|$. In this setting, assuming that $d < n^{1-\\kappa}$, then the lower bound given in Lemma 1 of the referenced paper is $O(n \\cdot  \\lambda/d^2) = O(n^{1-\\kappa}/d^2)$. This is loose by a $d^2$ factor, as compared to our tight lower bound.
>
> __"How does your upper bound on the sample complexity compare with that of "Coresets for Near Convex Functions"...with respect to the regularized loss function?""__
>
> The key difference is that our coreset sizes have no dependence on a regularization parameter -- they hold even for the unregularized loss function. However, they do depend on the complexity parameter mu(X). Thus, the results are not directly comparable. It seems that the main result for logistic regression in the referenced paper (Cor 7) is similar to the cited result of Curtin et al. in the regularized case, being worse by roughly a d factor. Thus, for small d, that upper bound is close to our lower bound for the regularized case.
>
> __"When does your approach outperform that of [MSSW18] (practically speaking)?"__
>
> Our $l_1$ lewis weight sampling and the square root $l_2$ leverage score sampling of MSSW18 are similar in intuition, and we did not see large deviations in their performance. Our $l_1$ lewis weight sampling however, gives a stronger theoretical result. It would be interesting if one could show that square root $l_2$ leverage score sampling gives a similar result, or else construct an example where it truly requires larger coreset sizes. However, it seems that in practical instances, such examples may not typically arise.
>
> __"It would be good to see a comparison between your approach and sensitivity-based approaches."__
>
> Yes -- this is a good suggestion. The results of Curtin et al. give tight results in the regularized setting via simple uniform sampling (which acts as sensitivity sampling). We compare to, and generally outperform, this method. However, it would be interesting to compare to the sensitivity estimation methods presented in the referenced papers above in both the regularized and unregularized setting.
>
> __"Can you plot the approximation error as a function of time?"__
>
> We avoided any reporting of runtime, using coreset size as a surrogate quantity, due to the variability in runtime depending on the actual implementation, architecture, optimization method, etc. However, we agree that having some preliminary runtime results would be helpful.

---

### Official Review · Reviewer_cpmn · 2021-07-07

**Rating:** 8
**Confidence:** 4

**Summary:**

The authors propose a novel way of building coresets for linear classification problems. They build upon the two following works:
- [MSSW18], in which the concept of the classification complexity measure $\mu_y(X)$ is proposed and first used in coreset theorems
- [CP15], in which normalized $\ell_p$ Lewis weights are shown to be a good probability distribution for row sampling of tall matrices $A\in\mathbb{R}^{n\times d}$, when one wishes to preserve the $p$-norm of the product $Ax$ for any vector $x\in\mathbb{R}^d$.

The authors show that $\ell_1$ Lewis weights of the input data matrix $X\in\mathbb{R}^{n\times d}$, are useful to build a good probability sampling distribution over the data elements, in order to create efficient coresets for linear classification. To do this, they:
- first show that normalized $\ell_1$ Lewis weights are a good probability distribution for sampling coresets when the classification's loss function is the ReLU function; as the proof is elegantly short in this case. This strategy guarantees coreset of size $\tilde{O}\left(d\mu(X)^2\right)$ whp. The proof is mainly based on  proof strategies found in [CP15].
- then, via the definition of so-called nice Hinge functions --basically functions that are similar enough to the ReLU, the authors generalize their results to the hinge loss or the log loss functions. The proof needs to separately take into account different subspaces of $\beta$'s depending on the value of $||X\beta||_1$.

Their coreset construction is an improvement (the guaranteed minimal coreset size is much smaller) over similar approaches that take into account this $\mu(X)$ parameter (basically, [MSSW18]).

The paper finishes with experimental comparisons with the state-of-the-art from [MSSW18] (that samples with a combination of uniform and square root of the classical $\ell_2$ leverage scores, an oblivious sketching algorithm, and also simple uniform sampling. The algorithm used to estimate the $\ell_1$ Lewis weights is that of [CP15].

**Main Review:**

This paper is well written, and I could not find any error in the proofs. The subject is timely and interesting. The true contribution of the authors, as I see it, is to combine ideas from [MSSW18], [CP15]; as the proofs are mainly based on proof techniques from [CP15]. The exception is the extension to nice hinge functions, for which the authors contributed a way to extend the results easily obtained on the ReLU to other similar hinge functions.

This paper should be accepted.

Minor:
- beware of the indices of your sums. There are some places where a $n$ should be a $m$ (equation of Thm 3 for instance)
- please show why, when each $p_i$ is a scaling of a constant factor approximation of the Lewis weight, $S$ has that number of rows. I find that the sentence "scaling of a constant factor approximation of the Lewis weight" could be clearer and showing with a few calculations what you mean will make it clearer.

**Time Spent Reviewing:**

6

---

> ### Author Response · Authors · 2021-08-09
> **We clarify the coreset size claim and precise meaning of 'scaling of a constant factor approximation...'.**
>
> Thanks for the positive review. Thanks for pointing out the index typos -- we will resolve.
>
> We will also clarify the coreset size claim and the precise meaning of _'scaling of a constant factor approximation...'_. We have $\sum_{i=1}^n \tau_i(X)  = d$, which gives that $m = \\sum p_i = c \\cdot d \\cdot \\log(m/\\delta)/\epsilon^2$ for some constant $c$. If we set $m = O(d \\cdot \\log(d/(\\delta \\cdot \\epsilon))/ \\epsilon^2)$ we have $\\log(m/\\delta) = O(\\log(d/(\\delta \\cdot \\epsilon))$ which gives $m = c \\cdot d \\cdot \\log(m/\\delta)/\\epsilon^2$ as needed.
>
> Note that there is a typo in  the value for $m$ in Theorem 3 -- there is a $\\log(d/\\epsilon)$ in place of $\\log(d/(\\delta \\cdot \\epsilon))$. However, it is written correctly in Theorem 5 and all following results. We will correct that.

---

> > ### Comment · Reviewer_cpmn · 2021-08-23
> > **thank you for the clarification**
> >
> > thank you for the clarification

---

### Official Review · Reviewer_uSXH · 2021-07-14

**Rating:** 8
**Confidence:** 4

**Summary:**

This paper provides improved coresets for a wide class of linear classification problems, improving the seminal work of [MSSW18]. In particular, it improves the d^3 \mu^3 / \epsilon^4 bound of [MSSW18] to d \mu^2 / \epsilon^2, which is significant. The parameter \mu measures the imbalance of the data, and it was introduced by [MSSW18] and shown to be necessary for a small-size coreset to exist.

Technically, the improvement is achieved by using L_1 Lewis sampling which is different from the standard Feldman-Langberg approach commonly used in coreset literature. The experiments show that the new coreset can largely outperform the uniform baseline, and is overall comparable with [MSSW18] while slightly outperform it when \mu is large.


**Limitations And Societal Impact:**

Yes.

**Main Review:**

This paper is interesting not only in that it presents an improved result, but also the techniques could have value for future research. Indeed, as observed by the authors, coresets based on the Feldman-Langberg framework has size at least d^2, where one factor of d is from the sensitivity and the other is from the shattering dimension. It's nice to get rid of such dependence. Actually, in other important problems that admit coresets such as k-means, such phenomenon is also observed. For k-means problem, the state-of-the-art coreset size is at least k^2/eps^4, where k is the sensitivity and k/eps^2 is the dimension (of JL). This paper's technique may suggest a way to bypass this limitation of the framework.

**Time Spent Reviewing:**

4

---

> ### Author Response · Authors · 2021-08-09
> **Thanks very much for the positive review.**
>
>  We agree that using Lewis weight based techniques might be an interesting approach to generally improving coreset sizes for different problems, by avoiding the direct union bound that typically arises in sensitivity-based approaches.

---

### Official Review · Reviewer_mLvM · 2021-07-15

**Rating:** 7
**Confidence:** 4

**Summary:**

This paper studies construction of coresets for training linear classifiers with objective functions such as logistic loss and hinge loss. The coresets they construct are of size $d\mu_y(X)^2/\epsilon^2$ upto polylogarithmic factors where $\mu_y$ is the parameter that denotes the maximum ratio of the correctly classified points to the wrongly classified points by a linear classifier. The authors argue that in general as all the points in the example cannot be classified by a linear classifier exactly, the parameter $\mu_y(X)$ is low and therefore  can have small coresets. For constructing the coreset, the authors sample points independently from a distribution based on $\ell_1$ lewis weights which were earlier used to construct $\ell_1$ subspace embeddings with small number of rows.

**Main Review:**

Strengths:
- Applying lewis weights to construct a coreset for a new problem.
- A single sampling distribution works for many types of losses.
- Paper writing is good.

Weaknesses:
- The paper lowers the exponents from earlier constructions but the results aren't very novel. Unless coupled with a lowerbound that matches these parameters, the results in this paper aren't very interesting.
- My issue with the parameter $\mu_y(X)$ is that if it is small, then maybe linear classifiers aren't good for that dataset at all. So why would you even want to find a linear classifier where even the best linear classifier performs poorly. Also, when the parameter $\mu_y(X)$ is small, it seems like a coreset construction algorithm may not have to do much at all. It would be good to plot the fraction of points in the test set that are correctly classified by the $\tilde{\beta}$ computed using the coreset vs the number of points in the test set that are correctly classified by a $\beta$ computed by using a uniform sample of the same size as the coreset. I believe this would give a much better picture of how these coresets improve upon a uniform sample.

Post Rebuttal:
I'd like to increase my score mainly because they have a near-linear dependence in $d$ and I now realize this may not be possible using earlier techniques and therefore the techniques in this paper may be helpful to decrease coreset sizes for other problems.

**Time Spent Reviewing:**

4

---

> ### Author Response · Authors · 2021-08-09
> **We report classification accuracy for various methods.**
>
> Thanks for reviewing our paper and for suggesting that classification accuracy of the classifiers learnt using various methods is a useful metric. We report the AUC values on a holdout set for various methods and datasets, and for different sizes of the coresets. The reported numbers are median values over 40 runs.
>
> We observe the following things from the experimental results.
> 1.	For Webbspam and Kddcup datasets, lewis (our method) and l2s have higher AUC values as compared to uniform sampling. This especially true for small coreset sizes.
> 2.	For Covertype, all methods are very close to each other. This is especially clear when we plot the error bars as they are highly overlapping.
>
> OpenReview doesn’t let us attach plots, so we are reporting the numbers as a table.
>
> $$
> \\begin{array}{|c||c|c|c||c|c|c||c|c|c|}
> & Covertype & Covertype & Covertype & Webbspam & Webbspam & Webbspam & Kddcup & Kddcup & Kddcup\\\\
> size &  uniform &   l2s &  lewis &  uniform &   l2s &  lewis &  uniform &   l2s &  lewis \\\\
> \\hline
> 50    &     0.606 &  0.616 &   0.599 &     0.817 &  0.843 &   0.875 &     0.956 &  0.983 &   0.993 \\\\
> 100   &     0.657 &  0.658 &   0.637 &     0.856 &  0.888 &   0.904 &     0.961 &  0.993 &   0.996 \\\\
> 200   &     0.687 &  0.694 &   0.682 &     0.892 &  0.918 &   0.921 &     0.974 &  0.996 &   0.997 \\\\
> 500   &     0.705 &  0.713 &   0.711 &     0.909 &  0.944 &   0.942 &     0.987 &  0.998 &   0.998 \\\\
> 1000  &     0.710 &  0.721 &   0.716 &     0.934 &  0.960 &   0.957 &     0.992 &  0.998 &   0.998 \\\\
> 2000  &     0.730 &  0.734 &   0.729 &     0.954 &  0.967 &   0.966 &     0.995 &  0.998 &   0.998 \\\\
> 5000  &     0.730 &  0.745 &   0.735 &     0.969 &  0.971 &   0.970 &     0.998 &  0.998 &   0.998 \\\\
> 10000 &     0.759 &  0.756 &   0.740 &     0.971 &  0.972 &   0.972 &     0.999 &  0.998 &   0.998 \\\\
> \\end{array}
> $$

---

### Decision · Program_Chairs · 2021-09-27

**Decision:**

Accept (Poster)

**Comment:**

After the rebuttal phase, all reviewers see the merits of the paper and this coincides with
my own impressions. The paper is worth being published.